# Laying Performance, Egg Quality Characteristics, and Egg Yolk Fatty Acids Profile in Layer Hens Housed with Free Access to Chicory- and/or White Clover-Vegetated or Non-Vegetated Areas

**DOI:** 10.3390/ani11061708

**Published:** 2021-06-07

**Authors:** Canan Kop-Bozbay, Ahmet Akdag, Ayfer Bozkurt-Kiraz, Merve Gore, Orhan Kurt, Nuh Ocak

**Affiliations:** 1Department of Animal Science, Faculty of Agriculture, Eskisehir Osmangazi University, 26480 Eskisehir, Turkey; cbozbay@ogu.edu.tr; 2Department of Animal Science, Faculty of Agriculture, Ondokuz Mayis University, 55139 Samsun, Turkey; nuhocak@omu.edu.tr; 3Department of Animal Science, Faculty of Agriculture, Harran University, 63290 Sanliurfa, Turkey; ayferbozkurtkiraz78@gmail.com; 4Department of Field Crops, Faculty of Agriculture, Ondokuz Mayis University, 55139 Samsun, Turkey; mervegore89@gmail.com (M.G.); orhank@omu.edu.tr (O.K.)

**Keywords:** free range, egg production, unsaturated fatty acid, cholesterol, herbage intake

## Abstract

**Simple Summary:**

Outdoor-based systems can improve the product quantity and quality in laying hens. This study investigated the laying performance and several egg quality characteristics in layer hens fed on a conventional diet with free access to a soil area (control, C), a chicory (CI)- or white clover (TR)-vegetated area, or a CI and TR mixture (MIX)-vegetated area. The C hens consumed more concentrate feed, without affecting the laying rate, than did TR and MIX hens. Herbage intake (HI) of the TR and MIX hens was higher than that of the CI birds. The C hens produced eggs with thicker shells than the CI, TR, and MIX hens. The decrease in the feed intake and the improvement in fatty acid (FA) profiles of the egg yolk was related to the HI. Concerning the TR and MIX vegetation, the FA composition of herbage contributed to the production of eggs with preferred FA attributes, such as polyunsaturated FAs and a favourable *n*-6 to *n*-3 ratio.

**Abstract:**

This study investigated the laying performance, egg quality, and egg yolk fatty acids (FAs) and cholesterol content in layer hens housed with free access to chicory- and/or white clover-vegetated areas. During a 16-week study, 400 Lohmann Brown hens (32 weeks old) housed with free outdoor access were allocated randomly into four groups, each with four replicates of 25 hens. Control hens were fed a conventional diet with free access to a soil area (C), whereas other hens were fed on a conventional diet with free access to a chicory (CI)- or white clover (TR)-vegetated area or a CI and TR mixture (MIX)-vegetated area. The C hens consumed more concentrate feed (*p* = 0.018) than the TR and MIX hens, which had a higher herbage intake than the CI birds (*p* < 0.001). The C hens produced eggs with a thicker shell than those in the other treatment groups (*p* = 0.013). Compared with C, the saturated FAs of egg yolk decreased for MIX (*p* = 0.010). The polyunsaturated FAs were higher in the MIX eggs than in the C and TR eggs (*p* < 0.001). Although FAs were distributed in all quadrants of the principal component analysis (PCA), three main FA profiles could be identified based on the loadings of natural groupings in the PC2 versus PC1 plot. The present study shows clear evidence for the contribution of herbage to the hen diet without affecting laying performance. In addition, the FA composition of the CI and MIX vegetation contributed to the production of eggs with preferred FA attributes, such as polyunsaturated FAs and a favourable *n*-6 to *n*-3 ratio.

## 1. Introduction

In recent years, there has been an increased consumer demand for meat and egg products that focuses on animal welfare during production and product safety and quality [1,2,3,4]. As such, feeding forage or pasture intake in outdoor-based systems (conventional free-range systems or organic systems) for laying hens has gained tremendous attention in the poultry research field, as highlighted by certain reviews [2,3,4,5,6,7]. A considerable body of research has shown that in general, these systems delivered low egg production and a high feed conversion ratio (FCR), but they delivered high-quality and heavier eggs compared with conventional systems [1,8,9,10,11,12,13,14,15,16]. These findings resulted from reduced feed intake and diluted diets due to forage or pasture intake versus increased energy requirements of laying hens [3,17]. Moreover, information about feed intake from the outdoor areas is still scarce. There is no clear evidence that laying performance and egg quality properties are improved when hens are raised with free access to outdoor areas with vegetation, as has been reported previously [11,18]. Therefore, further studies are warranted.

White clover (*Trifolium repens* L.) is a stoloniferous perennial plant capable of symbiotically fixing atmospheric nitrogen, resulting in high protein content and high nutritive value [4,19]. The chicory (*Cichorium intybus* L) plant, a small perennial herbaceous aromatic herb, contains nutrients (carbohydrates, proteins, vitamins, and minerals) and bioactive phenolic compounds (inulin, esculin, terpenes, coumarins, and flavonoids) [20]. These swards can increase herbage production and further improve herbage nutritive value [19] and thus can contribute to a more sustainable production system. When hens are safely grazed on an area with white clover or chicory plants, the dietary amounts consumed during free-ranging contribute to the total feed intake to meet the nutrient needs of hens [21]. These plants can also positively impact hen health and product quality [4,20] because of the increased intake of bioactive substances [21].

Because free-range hens consume different levels of herbage in areas with various types of vegetation, their laying performances and egg qualities depend on herbage intake (HI) and forage species [8,9,10,11,16]. Different herbage species, including white clover or chicory species, have been used in outdoor-based production systems of laying hens, but with contradictory results regarding laying rate and certain internal and external egg quality properties [1,3,4,10,16]. Despite these observations, little information is available regarding the effects of the chicory and white clover species, used either as a single herb or a mixture in vegetated areas, on the laying performance and certain egg quality characteristics in layer hens fed a conventional diet. Fatty acid (FA) enrichment of the egg yolk is a key means to improve the nutraceutical properties of eggs [22]. Some herbage species (grass, grass-clover, alfalfa, or chicory) consumed by free-range hens have enriched the FA content of egg yolk [4,9,14,16] depending upon the range utilization ratio [4,10]. However, limited studies related to outdoor-based systems of laying hens have focused on the association between FAs of the egg yolk resulting from FAs ingested via the HI [4,16].

It is possible that free access to chicory- and/or white clover-vegetated areas may improve egg production, feed intake, FCR, and egg quality characteristics (including the FA profiles, cholesterol content, and colour of the egg yolk) in layer hens fed with a conventional diet. In addition, it can provide knowledge about how the FA profiles of these species (as a single herb or a mixture) are distributed between the FAs of the egg yolks. Therefore, this study was designed (I) to test for distinct differences in the laying performance and egg quality characteristics in layer hens fed on a conventional diet with free access to the outdoor area, either vegetated or unvegetated with the chicory and/or white clover species, and (II) to characterize changes in the FA profile of the egg yolk produced by hens fed on a conventional diet with free access to the outdoor area to better understand the regulation process of FA transfer from the consumed herbage to the egg yolk. Accordingly, the main aims were to (I) compare the effects of free access to chicory- and/or white clover-vegetated areas or non-vegetated soil areas on the laying performance of layer hens fed with a conventional diet and on several egg quality traits, (II) to evaluate the reorganization of yolk FAs following the free access to outdoor areas with or without vegetation, and (III) to more clearly explain the contribution of the HI to the production of eggs with preferred FA attributes using a chemometric approach.

## 2. Materials and Methods

This study was carried out from June to September 2019 using the poultry facilities of the Faculty of Agriculture, Eskisehir Osmangazi University, Eskisehir, Turkey (39°45′42″ N and 30°28′40″ E, 813 m above sea level). In Eskisehir, the climate is warm and temperate, with an average annual rainfall of 393 mm, a dry period of three to four months and an annual average temperature of 10.9 °C (max 21.7 °C, min −0.1 °C).

### 2.1. Hens, Housing, and Feeding

After a pre-trial period (two weeks) of adaptation to the environment and diet, 400 Lohmann Brown layer hens (1777 ± 4.75 g mean body weight with 92.7% flock uniformity) aged 32 weeks were allocated into four groups with four replicate floor pens of 25 hens each. Body weight and egg production were measured during the pre-trial period, and the hens in each of the replicate pens were homogeneous in terms of these variables. All hens were vaccinated against Marek and Newcastle diseases (Nobilis^®^ MSD Animal Health). The experimental period lasted 16 weeks. All hens were raised in floor-litter pens in a curtain-sided house with outdoor access during daylight hours, which was ventilated both naturally and mechanically and illuminated both artificially via white LED lightbulbs and naturally through the windows. Each of the indoor pens (2.5 × 3 m) was equipped with a perch, individual nests (30 × 45 × 60 cm height, 1 nest/5 hens), an automatic drinker, and a red circular poultry feeder plate. Drinkers were also available in each outdoor plot (2.5 × 10 m). Indoor pens with outdoor access in the same building were randomly assigned for each treatment. At the side of the building, there was one 32 × 32 cm doorway (pop holes) from each indoor pen for access to its own outdoor plot. During the experiment, the indoor temperature was set at 20 ± 1 °C. The hens were exposed to artificial plus natural light, depending on the daily photoperiod, to provide light for 16 h (16 h light: 8 h dark) and confined to indoor pens at night.

A corn–soybean meal-based conventional diet in mash form and water were provided ad libitum throughout the experimental period for all treatments. The ingredient and chemical composition of the experimental diet are presented in Table 1.

Control hens freely accessed a non-vegetated soil area (C), whereas other hens freely accessed chicory (CI)-, white clover (TR)-, or mixed (MIX)-vegetated areas. In the MIX group, the chicory and white clover were mixed in a 1:1 ratio. Feeding (with the exception of outdoor vegetation species), rearing conditions (temperature, photoperiod), and prophylaxis procedures (https://www.lohmanngb.co.uk accessed on 5 June 2019) were the same for all groups until the end of the experimental period.

### 2.2. Data Collection, Measurements, and Analysis

Hens were weighed at placement and after 16 weeks of the experiment. During the experiment, feed intake (FI) was recorded at 14-day intervals, while all eggs, their individual weights, and the number of defective eggs (broken, cracked, or without eggshell) were recorded daily per replicate pen. Egg-laying rate (%), egg mass (laying rate × egg weight), FI (total FI/number of days of the trial period) and FCR (g feed: g egg mass) were calculated. Of the eggs produced during the last 3 days of each 14-day interval, 64 randomly selected eggs (4 from each replicate) were used to determine certain internal (albumen and yolk percentages, Haugh unit, and yolk colour score) and external (eggshell weight, thickness, and breaking strength) quality parameters [23]. Egg weight was measured on a digital scale with accuracy to the nearest 0.001 g. The egg yolk, egg white (albumen), and eggshell of the cracked egg were weighed on the same digital scale and then the proportions of yolk ((yolk weight/egg weight) × 100), albumen ((albumen weight/egg weight) × 100), and shell ((shell weight/egg weight) × 100) in the egg were calculated. The egg shape index value ((width/height) × 100) was calculated using the height and width values of the egg measured with an electronic calliper. To calculate the egg yolk index ((height/diameter) × 100) and albumen index ((height/(length + width)) × 100), the height, width. and length of both the yolk and albumen were measured with a tripod micrometre and an electronic calliper, respectively. Haugh unit score was calculated using the egg weight and albumen height (albumen height + 7.57 − 1.7 × egg weight^0.37^). The eggshell thickness was determined at three random locations of the eggshell that represented the whole surface of the egg using a micrometre. The breaking strength of the eggshell (kg/cm^2^) was measured using a breaking strength measuring device (egg force reader) that applied pressure on the pointed end of the egg with a screw [23]. To distinguish the yolk colour density, the DSM Yolk Colour Fan with a 16-scale colour index was used.

The outdoor plots, divided by woody fences into two sub-plots, were rotationally grazed at 14-day intervals. The HI in outdoor replicate plots was estimated according to the sward cutting technique, using a metallic frame (50 cm × 50 cm) at 14-day intervals with a fixed cutting height of 2 cm [24]. Accordingly, the HI was calculated using the herbage mass present at the introduction of the hens in each plot, the herbage that remained at the end of the next interval, and the undisturbed mass of grass from ungrazed areas within the metallic frame [9]. The herbage samples were collected from the vegetation within the metallic frame in each of the plots.

The chemical composition (dry matter, crude protein, ash, neutral detergent fibre and acid detergent fibre contents) of herbage (Table 2) was analysed using the approved methods [25]. The FA profiles of the feed, forage species, and egg yolks were determined by gas chromatography (GC) analysis after lipid extraction [16,26] at the end of the experiment, at 28-day intervals, and at 5-week intervals, respectively. The GC analysis was performed on a GC-2010 Pro Capillary Gas Chromatograph equipped with a 20 m length, 0.1 mm film, 0.1 μm internal diameter capillary column (Shimadzu Corp., Kyoto, Japan), and nitrogen as a carrier gas. The FA profiles of the samples were identified using GC solution software to compare their mass spectra and retention time peaks. The FA values were expressed as weight percentages (% of total FAs). The samples for each FA were analysed in duplicate. The FA profiles of forage samples are given in Table 3. The cholesterol content (mg/dl) of the egg yolk was analysed at five-week intervals. For cholesterol analysis [23,27], two eggs from each replicate pen (eight eggs per treatment) were weighed and boiled for 5 min. Then, the yolks separated from the albumens were homogenized in a vortex and dissolved in isopropanol (4 mL/0.1 g of yolk). These samples were centrifuged at 3000 rpm for 5 min and kept for 10 min in a 37 °C water bath. The cholesterol contents (mg/dl) of the filtered samples were determined by the Epoch Microplate Spectrophotometer (BioTek Instruments, Inc., Winooski, VT, USA) using a cholesterol assay kit (CS0005 Sigma-Aldrich, Taufkirchen, Germany).

### 2.3. Statistical Analysis

For all data, the replicate pen served as the experimental unit. For normality and homoscedasticity, all data were verified by the Kolmogorov-Smirnov test and Levene’s test, respectively. Data expressed as percentages were subjected to arcsine transformation prior to analysis to normalize the distribution of residuals; however, actual percentages are reported. Data were subjected to one-way ANOVA using SPSS software (Version 21.0, SPSS, Chicago, IL, USA). The differences among the means were deemed significant at *p ≤* 0.05, according to Duncan’s multiple range test. To understand the impact of each FA within the forage species and all the variables considered simultaneously, principal component analysis (PCA) was applied. The means of four repetitions of each group at five-week intervals (*n* = 96, four treatments × four replicate pens × two eggs × three sampling intervals) were used as the cases. Before performing PCA, the suitability of data for factor analysis was assumed using the Kaiser-Meyer-Olkin (KMO) test and Bartlett’s test (KMO = 0.639; χ^2^ = 941.8, *p* < 0.001). Thus, a new set of 11 orthogonal variables was generated by PCA. Only the principal components (PCs) that had eigenvalues of > 1.0 were considered significant to describe most of the total data variations [28].

## 3. Results

No differences were found in the final body weight, laying rate, or egg weight (Table 4). The C hens consumed more concentrate feed (*p* < 0.05) and had a numerically higher FCR compared with the TR and MIX hens. The HIs of TR and MIX hens were higher than that of the CI birds (*p* < 0.05). This HI amount numerically improved the FCR of the TR and MIX hens. The CI, TR, and MIX treatments did not affect either internal or internal egg quality characteristics, except for eggshell thickness (Table 5). The C hens produced eggs with a thicker shell than the CI, TR, and MIX hens (*p* < 0.05). However, this difference in the eggshell was not reflected in the cracked or broken egg ratio (Table 4).

In total, 11 FAs containing two trace FAs (> 0.1% weight) were identified. These two FAs, lauric acid and myristic acid, are not shown in Table 6. Significant differences were found among treatments in terms of both total FAs, except for *n*-3 FAs, and individual FAs, except for linoleic acid and linolenic acid (Table 6). Among the treatments, the lowest value of palmitic acid was recorded for the MIX group (*p* < 0.05). The TR and MIX hens produced yolk with higher stearic acid than the C and CI hens (*p* < 0.05). The behenic acid content of the yolk was lower in the eggs from the CI hens than those from the C and TR birds (*p* < 0.05). The eggs from the MIX hens displayed lower arachidic acid content compared with the other treatments (*p* < 0.05). In the samples of the C eggs, saturated FAs (SFA) constituted about 38.2% of the total amount of FAs. This percentage was at the same level for the CI and TR groups but was decreased for the MIX group (*p* < 0.05). The most abundant of the saturated FAs was palmitic acid, constituting about 26.8% of the total FA profile. The C hens produced yolk with higher palmitoleic acid levels compared with the TR and MIX hens (*p* < 0.05). The oleic acid content of the yolk was higher in the CI eggs than in those from the TR and MIX birds (*p* < 0.05). The C hens produced yolks that contained more monounsaturated fatty acids (MUFA) compared with the yolks of the TR hens (*p* < 0.05). The eggs from the C hens displayed higher arachidonic acid content compared with the CI, TR, and MIX hens (*p* < 0.05). The polyunsaturated fatty acid (PUFA) content of the yolks from the MIX hens was higher than that from the C and TR hens (*p* < 0.05). The corresponding value of the TR hens was higher than that of the C hens (*p* < 0.05). Among the PUFAs, the most abundant acid was linoleic acid, but it was not significantly different between all investigated groups (*p* = 0.155). The total *n*-6 FA content for eggs from the C hens was lower than that from the other treatment hens (*p* < 0.05). The *n*-6 to *n*-3 ratio for eggs from the CI hens was higher than that from the hens undergoing other treatments (*p* < 0.05). The corresponding ratios of the C and TR hens were higher than that of the MIX hens (*p* < 0.05). No differences were found in the cholesterol content of the egg yolk (*p* > 0.05).

The most significant PCs generated from the yolk FA data and their statistical loadings in the present study are depicted in Figure 1. The first and second PCs, i.e., PC1 and PC2, accounted for 70% of the variability in the data set from the four outdoor-based systems and had the highest eigenvalues, which were 6.44 and 1.35, describing 58.5% and 12.2% of the total variance, respectively (Figure 1a). Although FAs were distributed in all quadrants of the PCA, the loadings (or scores) corresponding to the PCs indicated high contributions from three groups. Therefore, from the PCA, three main FA profiles may be identified based on natural groupings in the PC2 versus PC1 plot. Group 1 was composed of FAs with positive loadings for PC1 and PC2 (stearic (0.950 and 0.133) and linolenic acids (0.707 and 0.596)). Group 2 included FAs with positive loadings for PC1 and negative loadings for PC2 (palmitic (0.989 and −0.039), behenic (0.908 and −0.062]), arachidonic (0.646 and −0.463), and lauric (0.586 and −0.281]) acids). Group 3 was composed of FAs with negative loadings for PC1 (oleic (−0.946 and −0.259), palmitoleic (−0.827 and −0.435), linoleic (−0.816 and 0.480), arachidic (−0.370 and 0.270), and myristic (−0.303 and 0.340) acid variables). Based on the correlation matrix loadings (≥ 0.75 and positive factor loadings) of the variables, stearic, palmitic, and behenic acids contributed most strongly to PC1, while linolenic, arachidonic, and lauric acids contributed less strongly.

Figure 1b depicts the score plots of FAs in the yolk from all treatments, generated by comparing the groups based on PC1 and PC2. The FAs of groups 1 and 2 were strongly related to the TR and MIX treatments, while group 3 was related to the C and CI treatments. The position of yolk samples in the score plot was consistent with the vegetation type in the outdoor area and also demonstrates that PC1 described the FA distribution between the studied herbages. The TR and MIX treatments were characterized by high values for stearic, palmitic, and behenic acids, while the CI group was characterized by high myristic and arachidic acids. The C group was somewhat in the middle and less tightly clustered than either of the other groups. Therefore, PC1 separated the TR and MIX samples from the CI samples. PC2 did not enable the C samples to be separated from the CI and MIX treatments, which were associated with FAs from groups 1 and 2, respectively.

## 4. Discussion

In the present study, free access to chicory- and/or white clover-vegetated areas decreased the concentrate FI of hens and contributed to the production of eggs with preferred FA attributes without affecting laying performance. These results did not confirm the suggestion that outdoor-based systems cause low egg production, high FCR, less weight uniformity, and high mortality compared with conventional production systems [1,8,9,10,12,13,14]. However, free access to permanent or temporary vegetated areas did not negatively affect laying performance [8,9,11] of the hens, despite increased egg and yolk weights [11,16]. In addition, some studies [9,11,16] indicated that vegetation type in outdoor areas was a potential contributor to nutrient content, including FAs, as well as certain quality parameters of eggs. These inconsistencies between our results and results reported previously may be attributed to environmental conditions, the strain and age of hens, the level of dilution of diets due to forage or pasture intake, grazed outdoor area, and the change in the energy requirements of hens depending on laying rate [3,8,11,17].

Although it is difficult to describe the supply of nutrients to the hen from the HI in outdoor-based systems [4], in general, the HI contributes to the nutrition of laying hens [3,4,21]. Laying hens can consume 2–57 g of dry matter per day from herbage [29,30], depending on the forage species or vegetation type [31], age and genotype of the hens, rearing conditions, etc. [30]. Although the hens with access to the TR- and MIX-vegetated areas consumed more herbage compared with those in the CI-vegetated area, the HI of hens in the present study were within this range. These differences between treatments may be related to the fact that the nutritive value of the outdoor area changed according to both vegetation type [32] and macro-invertebrates living in the outdoor area [4,10]. Indeed, the white clover, a leguminous species, and the mixture of chicory and white clover displayed relatively high crude protein and low neutral detergent fibre content compared with chicory, which belongs to the Asteraceae family. Unfortunately, the contribution of macro-invertebrates to the diet of hens was not quantified in our study.

The data relating to the final body weight suggests that despite a reduction in the concentrate feed intake, the HI promoted weight gain in the hens kept under the same rearing conditions. This may show that there were no severe effects of the change in digestibility due to fibre content of plant species on the performance of herbage-consuming hens, as Iqbal et al. also revealed [17,32]. Our result regarding the HI supports the idea that poultry with access to pasture often consume herbage in quantities resulting in diet dilution and reduced concentrate feed intake [3,32]. It is expected that the decreased concentrate feed intake of hens that access the vegetated area results in the production of fewer eggs. However, the present HI might be used to cover the production demands of the vegetated area versus the non-vegetated area. In addition, the forage consumed from TR- or MIX-vegetated areas caused a numerically lower FCR compared with that in the other treatments. Mohammed et al. [10] noted that when the additional consumed forage was considered, an increase in FI was used to cover the high production demands and was subsequently utilized for producing more eggs. As such, our results indicate that establishing the amount and the nutritional value of herbage to be ingested is very important to manipulate the nutritional quality of the egg [9].

The HI might have changed the mineral content [15] and the calcium to available phosphorus ratio of the diet [3,32] and reduced intestinal calcium uptake because feed ingredients from plant sources are inadequate in meeting the requirements for calcium and phosphorus [33]. Therefore, the decreased eggshell thickness in the vegetated area groups, a major concern of the egg industry, could be associated with the consumed forage. High eggshell thickness is associated with high eggshell breaking strength [17,34]. However, the decrease in eggshell thickness in the hens consuming herbage was not reflected in the eggshell strength and cracked egg ratio. This finding, which concurs with Samiullah et al. [35], indicates that the decrease in eggshell thickness was not severe enough to have an impact on this parameter. This means that similarly sellable eggs are available for all treatments. When using chicory and white clover in outdoor-based systems, the producers could add extra vitamin D to the conventional diet and provide a free-choice calcium source to overcome the eggshell issues.

The results regarding the studied egg quality properties support the findings of studies that indicated no negative effects of the production system on these properties [12,16]. The Haugh unit, a measure of egg protein quality based on the albumin height, and egg yolk colour, an important commercial characteristic, are both important properties of egg quality, along with other parameters such as eggshell thickness and strength. In contrast to results reported previously [4,15,17,35], the HI of hens did not have a significant effect on the Haugh unit and egg yolk colour. This discrepancy among the studies might be due to differences in the housing type (deep or floor litter, cage, or enrich cage), vegetation type (grass, legume, other forbs, or their mixture), the range utilization percentage, the environmental conditions, and the age and strain of hens used [14,17,35,36]. Differences in environmental temperature between the free-range and conventional systems resulted in differences in egg weight [5] and Haugh units [14]. Therefore, we observed a similar egg weight and Haugh unit among the treatments due to the similar environmental temperature because all hens had free access to the outdoors. This similarity in the egg weights was similarly reflected in internal egg quality parameters, as reported by Ketta and Tůmová [34].

Because the egg yolk colour depends mainly upon the diet, the differences in the yolk colour of eggs produced in an outdoor-based system are attributed to the herbage materials consumed by the hens [4,10,14,29,37]. Horsted et al. [29] reported that hens that consumed chicory leaves produced eggs with a darker yolk colour than those consuming a grass-clover pasture. Our results did not confirm these notions since our study failed to prove a positive effect of the herbage consumed by hens on the yolk colour. However, the results regarding egg yolk colour are consistent with those of Yilmaz Dikmen et al. [38], who found that the influence of the rearing system on the colour of the yolk was insignificant. These discrepancies may result from the chemical and physical properties of egg yolk, the production cycle of the hens, and the varieties and maturity stage of the herbages used [1,14,37]. Moreover, the HI levels in the present study were probably not high enough to significantly affect the egg yolk colour [4]. Our similar results, namely the similar yolk colour of eggs from hens with free access to the vegetated or unvegetated outdoor area, was probably due to hens in all groups receiving a conventional diet containing ingredients with high pigments [37].

The consumption of the white clover and chicory-clover mixture had a larger effect than either alone on the levels of MUFA and PUFA, which play a crucial role among unsaturated fatty acids, mainly due to their physiological functions [25]. The results of the PCA confirmed previous findings on the FA content of the yolk from eggs produced by feeding forage or pasture intake in outdoor-based systems [9,14,36]. Corrales-Retana et al. [22] suggested that any change in the FA profiles of the yolk due to the diet of the hens is mainly linked to the lipid fraction, which represents the reserve lipid component of the cell. Walczak et al. [25] noted that the metabolism of FAs is dependent on the relative cellular FA composition, rather than the absolute concentration. These observations may explain why the PCA grouped the FAs of the yolk differently and the mutual correlations between FA profiles of the egg yolk modified by the FAs ingested via the HI. Mierliță [15] noted that the FA profile of egg yolk was characterized by a relatively low SFA and high MUFA and PUFA proportions, as reported herein. Moreover, findings relating to the FA profiles of the yolk were similar to previous reports [15,22,39] in which the yolk FA contents were maximized when the relevant FA was higher in the diet, reflecting a higher concentration of FAs in fresh herbage. Although the PUFA content of chicory species (61.79%) was higher than that of white clover (37.68%), the eggs from CI and TR treatments had similar contents of PUFA (22.43% and 22.17%, respectively). This may be due to the lower chicory intake (13.7 g/hen/day) than white clover intake (18.0 g/hen/day), relatively low consumption of both herbages [1,4], and the differences in the quantity and digestibility of non-starch polysaccharides found in these herbages [32]. Hammershøj and Johansen [4] noted that a high HI (i.e., ≥ 50 g/hen/day) is estimated to be necessary for beneficial effects on egg qualities such as FA content and egg yolk colour. The egg yolk is a rich source of the *n*-3 and *n*-6 acid family, the amount of which depends mainly on the genotype and age of the laying hens [25]. These findings show a significant dietary effect, as also revealed previously [22,39,40], and explain why there were progressive changes in the *n*-6 to *n*-3 ratio in egg yolk with altered linolenic acid and arachidonic acid concentrations. The score plots generated between PC1 and PC2 segregated the free-range vegetation type based on FA composition. Indeed, PC1 indicates that FAs from the TR and MIX groups had a greater effect on the FA profiles of the egg yolks compared with the CI group. Our results indicate that the CI and MIX vegetations contributed to the production of eggs with preferred FA attributes because a low *n*-6 to *n*-3 ratio in the human diet is favourable [22,25]. As such, white clover, and especially a mixture of chicory and white clover, may be successfully employed by farmers to provide more information on the variation in vegetation types than is possible with experimental data alone.

The cholesterol content of eggs is affected by many factors, such as the productivity, strain and age of the hens, housing system, season, etc. [9,41,42]. Indeed, studies have found higher levels of cholesterol in eggs produced by hens kept under free-range conditions [9,41]. This was not confirmed in our study. However, the result related to the cholesterol level of the egg yolk is consistent with that of previous reports [15,36] in which the rearing system (in cages vs. under free-range conditions) did not affect the cholesterol content of the egg yolk. These inconsistencies may be related to the housing system, strain and age of the hens, vegetation type, and HI level in the outdoor area [4,9,42]. Moreover, all groups of our study had eggs with similar albumen and yolk proportions, which could imply a similar cholesterol level [10].

## 5. Conclusions

To conclude, the present study shows clear evidence for the contribution of herbage to the hen diet. PCA results indicate that white clover, and especially a mixture of chicory and white clover, could be used to feed the laying hens to maintain reasonably improved FA profiles in the yolk without detrimental effects on laying performance or egg quality. These results suggest that free access to white clover and a mixture of chicory and white clover not only improves the FCR in laying hens but also produces healthier eggs for consumers. This is due to lower SFA and higher PUFA percentages in the egg yolks compared with those of hens with free access to a non-vegetated area. Eggs from hens with free access to an area with vegetation are usually rich in beneficial components for human health, but distinguishing eggs sourced from such areas is difficult. Implementing this forage program within their own production system may be beneficial to individual poultry producers. Furthermore, to facilitate identifying the benefits of forage or pasture intake, further studies are warranted to evaluate whether macronutrients, FAs, vitamins, minerals, and bioactive phenolic compounds of the forage species can influence the corresponding profiles of the eggs.

## Figures and Tables

**Figure 1 animals-11-01708-f001:**
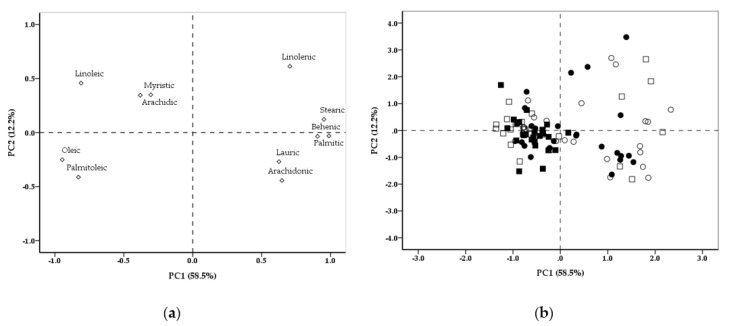
Loading plots (**a**) and score plots (**b**) of principal components (PC1 and PC2) for fatty acid profiles in the yolks from laying hens fed on a conventional diet and housed with free access to non-vegetated or chicory- and/or white clover-vegetated areas. ☐: C, housed with free access to non-vegetated soil area. ■: CI, housed with free access to chicory-vegetated area; ◯: TR, housed with free access to white clover-vegetated area; ●: MIX, housed with free access to chicory and white clover mixture-vegetated area.

**Table 1 animals-11-01708-t001:** The ingredient and chemical composition of experimental diet (as fed).

Ingredient	g/kg	Calculated Chemical Composition	%
Corn	580.0	ME (Mcal/kg)	2.75
Soybean meal (48% CP)	209.0	Crude protein	17.80
Sunflower meal (36% CP)	20.0	Ether extract	6.42
Full-fat soybean (37% CP)	70.0	Crude fibre	2.88
Limestone (38%)	95.0	Lysine	0.84
Dicalcium phosphate	19.0	Methionine + cysteine	0.69
Salt	2.5	Ash	11.12
Vitamin + mineral premix *	2.5	Calcium	4.10
DL-methionine	1.5	Available phosphorus	0.32
Lysine	0.5		

* Provided per kilogram of diet: vitamin A, 12,500 IU; vitamin D_3_, 2,500 ICU; vitamin E, 1.75 IU; vitamin K_3_, 4.5 mg; vitamin B_12_, 20 mg; riboflavin, 4.12 mg; D-pantothenic acid, 8.02 mg; folic acid: 750 mg; niacin, 19.8 mg; choline, 382.9 mg; Co, 100 mg; Cu, 5.0 mg; I, 100 mg; Fe, 50.35 mg; Mn, 64.26 mg; Se, 30 mg; Zn, 58.69 mg.

**Table 2 animals-11-01708-t002:** The nutrient contents of chicory (CI), white clover (TR), and their mixture in a ratio of 1:1 (MIX) used in the vegetated areas (% dry matter).

Nutrient	CI	TR	MIX
Dry matter	18.36	16.96	17.73
Crude protein	16.63	23.73	19.48
Ether extract	2.39	1.94	2.06
Acid detergent fibre	30.64	33.77	32.60
Neutral detergent fibre	47.66	35.16	42.58
Ash	0.93	0.98	0.97

**Table 3 animals-11-01708-t003:** Fatty acid (FA) composition of conventional diet (CD), chicory (CI), white clover (TR), and their mixture in a ratio of 1:1 (MIX) used in the vegetated areas.

FA (g/100 g FA)	CD	CI	TR	MIX
Myristic acid (C14:0)	0.05	0.52	0.47	0.50
Palmitic acid (C16:0)	12.31	20.88	27.01	24.84
Stearic acid (18:0)	2.08	5.30	7.42	6.48
Σ Saturated fatty acids	14.44	26.70	34.90	31.82
Palmitoleic acid (C16:1) *n*-7	0.13	1.51	3.04	2.92
Oleic acid (18:1) *n*-9	30.11	10.00	24.38	18.51
Σ Monounsaturated fatty acids	30.24	11.51	27.42	21.43
Linolenic acid (18:3) *n*-3	1.49	39.18	15.99	26.57
Linoleic acid (18:2) *n*-6	52.40	22.61	21.69	20.18
Σ Polyunsaturated fatty acids	53.89	61.79	37.68	46.75

**Table 4 animals-11-01708-t004:** Laying performance of laying hens fed on a conventional diet and housed with free access to non-vegetated or chicory- and/or white clover-vegetated areas.

Parameter ^1^	C	CI	TR	MIX	SEM	*p*-Value
Initial body weight (g/hen)	1792	1753	1775	1786	4.8	
Final body weight (g/hen)	1950	1927	1937	1936	10.3	0.282
Egg-laying rate (%)	95.8	94.5	94.5	95.2	1.25	0.673
Concentrate intake (g/hen/day)	130.4 ^a^	126.8 ^b^	124.5 ^c^	125.7 ^bc^	4.17	0.018
Herbage intake (g/hen/day)	-	13.7 ^c^	18.0 ^a^	15.5 ^b^	0.24	<0.001
Egg weight (g)	63.0	63.0	63.2	63.2	0.22	0.782
Egg mass (g/hen/day)	60.4	59.5	59.7	60.1	0.18	0.328
FCR (g feed: g egg mass)	2.16	2.13	2.08	2.09	0.011	0.195
Cracked egg ratio (%)	1.79	1.73	2.39	2.90	0.766	0.677

Treatment: C, housed with free access to non-vegetated soil area; CI, housed with free access to chicory-vegetated area; TR, housed with free access to white clover-vegetated area; MIX, housed with free access to chicory and white clover mixture-vegetated area. Abbreviations: FCR, feed conversion ratio; SEM, standard error of the mean. ^a,b,c^ Within a row, means with different superscripts differ significantly (*p* < 0.05). ^1^ Means represent four pens of 25 layers per treatment at 14-day intervals.

**Table 5 animals-11-01708-t005:** Egg quality traits of laying hens fed on a conventional diet and housed with free access to non-vegetated or chicory- and/or white clover-vegetated areas.

Parameter ^1^	C	CI	TR	MIX	SEM	*p*-Value
Haugh units	80.24	81.92	82.68	80.40	1.047	0.825
Yolk ratio (g/100 g egg)	27.29	27.54	28.37	27.97	0.243	0.695
Albumen ratio (g/100 g egg)	58.49	58.62	57.97	57.93	0.300	0.793
Eggshell ratio (g/100 g egg)	13.70	13.82	13.63	14.09	0.147	0.728
Yolk/albumen ratio	0.47	0.47	0.49	0.48	0.006	0.756
Eggshell strength (kg/cm^2^)	4.01	4.12	3.89	3.23	0.249	0.616
Eggshell thickness (μm)	0.46 ^a^	0.40 ^b^	0.41 ^b^	0.41 ^b^	0.001	0.013
Albumen index	8.75	9.05	8.69	8.35	0.321	0.900
Yolk index	44.17	44.25	43.67	48.22	0.748	0.094
Shape index	77.09	76.77	77.28	79.06	0.503	0.394
Yolk colour (grade 1–16)	12.1	12.0	11.8	12.0	0.095	0.583

Treatment: C, housed with free access to non-vegetated soil area; CI, housed with free access to chicory-vegetated area; TR, housed with free access to white clover-vegetated area; MIX, housed with free access to chicory and white clover mixture-vegetated area. ^a,b^ Within a row, means with different superscripts differ significantly (*p* < 0.05). ^1^ Means represent 16 eggs per treatment at 14-day intervals.

**Table 6 animals-11-01708-t006:** Fatty acid (FA) composition (g/100 g FA) and cholesterol content (mg/dL) of egg yolks from laying hens fed on a conventional diet and housed with free access to non-vegetated or chicory- and/or white clover-vegetated areas.

Variables ^1^	C	CI	TR	MIX	SEM	*p*-Value
Palmitic acid (16:0)	26.76 ^a^	25.74 ^b^	27.70 ^a^	27.39 ^a^	0.180	<0.001
Stearic acid (18:0)	10.31 ^b^	9.92 ^b^	12.04 ^a^	11.58 ^a^	0.226	0.001
Behenic acid (22:0)	0.26 ^a^	0.02 ^b^	0.10 ^b^	0.07 ^b^	0.021	<0.001
Arachidic acid (20:0)	0.90 ^a^	0.94 ^a^	0.91 ^a^	0.79 ^b^	0.014	0.001
Σ SFA	38.17 ^a^	37.09 ^ab^	37.92 ^a^	36.39 ^b^	0.213	0.010
Palmitoleic acid (16:1)	2.68 ^a^	2.65 ^ab^	2.33 ^c^	2.42 ^bc^	0.046	0.014
Oleic acid (18:1)	37.17 ^ab^	38.29 ^a^	35.28 ^c^	36.19 ^bc^	0.272	<0.001
Σ MUFA	41.12 ^a^	40.49 ^ab^	39.92 ^b^	40.61 ^ab^	0.135	0.017
Linoleic acid (18:2)	19.35	21.10	20.79	21.52	0.178	0.155
Linolenic acid (18:3)	1.22	1.28	1.31	1.48	0.054	0.191
Arachidonic acid (20:4)	0.13 ^a^	0.05 ^b^	0.06 ^b^	0.03 ^b^	0.011	0.006
Σ PUFA	20.72 ^c^	22.43 ^ab^	22.17 ^b^	23.05 ^a^	0.156	<0.001
Σ *n*-3	1.22	1.28	1.31	1.48	0.054	0.191
Σ *n*-6	19.48 ^b^	21.15 ^a^	20.85 ^a^	21.55 ^a^	0.152	<0.001
*n*-6/*n*-3	15.91 ^b^	16.55 ^a^	15.90 ^b^	14.56 ^c^	0.110	<0.001
Cholesterol	225.0	226.0	230.4	237.4	2.44	0.256

Treatment: C, housed with free access to non-vegetated soil area; CI, housed with free access to chicory-vegetated area; TR, housed with free access to white clover-vegetated area; MIX, housed with free access to chicory and white clover mixture-vegetated area. Abbreviations: SFA, saturated fatty acids; MUFA, monounsaturated fatty acids; PUFA, polyunsaturated fatty acids; SEM, standard error of the mean. ^a,b,c^ Within a row, means with different superscripts differ significantly (*p* < 0.05). ^1^ Means represent 16 eggs per treatment at 5-week intervals.

## Data Availability

The data presented in this study are available on request from the corresponding author.

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
