# Peer review of "Laying Performance, Egg Quality Characteristics, and Egg Yolk Fatty Acids Profile in Layer Hens Housed with Free Access to Chicory- and/or White Clover-Vegetated or Non-Vegetated Areas"

_animals, 2021, doi:10.3390/ani11061708_

Round 1

Reviewer 1 Report

  1. Table 2  no dry matter content that is not consistent with L172-173.
  2. Table 5 the yolk colors  is no differences among groups that is not reasonable for the three groups with HI intake.    

Author Response

Point 1: Table 2  no dry matter content that is not consistent with L172-173.

Response 1: Thank you for your determination. The dry matter contents of the herbages have been inserted into Table 2 in the second revised version of the MS.

Point 2: Table 5 the yolk colors is no differences among groups that is not reasonable for the three groups with HI intake.

Response 2: This is an important comment for us. As seen in the cited literature, it did not observe that herbage or pasture always affected the egg yolk. To avoid misunderstanding, this finding has been discussed again in a more concise form (Please check line 377 to 388 in the RVMS). 

Reviewer 2 Report

The Authors have answered all my main issues which were raised during first review, thank you. However, the revised version still requires clarifications. My concerns relate to the data presented in Fig 1. How it is possible, that after performing PCA for new data (96 data appoints (observations) instead of 12, score plot, fig 1b) you obtained the same distribution of loadings (fig 1a) ? In fact, it is completely improbable. Please revise - it must be clarified.

Minor comment: in fig axes labeling, correct commas to dots.

Author Response

Point 1: The Authors have answered all my main issues which were raised during first review, thank you. However, the revised version still requires clarifications. My concerns relate to the data presented in Fig 1. How it is possible, that after performing PCA for new data (96 data appoints (observations) instead of 12, score plot, fig 1b) you obtained the same distribution of loadings (fig 1a) ? In fact, it is completely improbable. Please revise - it must be clarified.

Response 1: Thank you for your determination because these are an important comment and suggestion for us. However, we disagree with you due to also accept by other referees. As stated in the response letter (coverletter.v1.pdf) to the referees for the first revision (did you see it? I don't know), we performed PCA for our 96 FA data. However, based on your comments, we saw that Figure 1b is not represented correctly. In the original MS, while figure 1a was prepared using 96 data, Figure 1b has been drawn using only 12 data because we thought it was sufficient only for the four replication dots of each treatment. Indeed, Figure 1a and Figure 1b have been drawn using the same 96 data appoints in the revised version of the MS.

Point 2: Minor comment: in fig axes labeling, correct commas to dots.

Response 2: Done as requested (Please check Figure 1 in the RVMS). 

Reviewer 3 Report

This revised manuscript is much improved.  Some minor changes are needed.

Line 78:  Finish the sentence:   “Fatty acid (FA) enrichment of the egg yolk with ( with what?) is a key means to improve the nutraceutical properties of eggs.

Line 172:  Change “cure protein” to ‘crude protein”

Lines 174:  remove the word “analyzed”

Line 177:  Change “was performed on GC-2010” to “was performed on a GC-2010”  (Add the word a)

Line 185:  change “albumens” to “albumen”  (remove the “s”)

Line 356:  change “used” to “using”

Line 379: change “consumed” to “consuming”

Lines 385-387:  Fix this sentence.  I think you mean to say something like this:

“The consumption of the white clover and chicory-clover mixture was more effective than either alone on the levels of MUFA and PUFA, which play a crucial role among unsaturated fatty acids, mainly due to their physiological functions.”

Line 412-414:  Change “Indeed, PC1 indicate that FAs  from the TR and MIX groups had a high effect on the FA profiles of the egg yolks compared to the CI group.”  To “ Indeed, PC1 indicates that FAs from the TR and MIX groups had a greater effect on the FA profiles of the egg yolks compared to the CI group.

Line 415: Change  “because the low n-6/n-3 ratio”  to “because a low n-6/n-3 ratio”

Lines 439-440:  Change “These results can be beneficial for their own production system of individual poultry producers.”  To “Implementing this forage program within their own production system may be beneficial to individual poultry producers.”

Author Response

Point 1: This revised manuscript is much improved.  Some minor changes are needed.

Response 1: Thank you for your important comment and suggestions.

Point 2: Line 78:  Finish the sentence:   “Fatty acid (FA) enrichment of the egg yolk with ( with what?) is a key means to improve the nutraceutical properties of eggs.

Response 2: The word "with" entered the text incorrectly during the first revision. Indeed, there is no that word in the original MS. Thus, this word has now been deleted.

Point 3: Line 172:  Change “cure protein” to ‘crude protein”

Response 3: Done as requested (Please check Figure 1 in the RVMS). 

Point 4: Lines 174:  remove the word “analyzed”

Response 4: Done as requested (Please check Figure 1 in the RVMS). 

Point 5: Line 177:  Change “was performed on GC-2010” to “was performed on a GC-2010”  (Add the word a)

Response 5: Done as requested (Please check Figure 1 in the RVMS). 

Point 6: Line 185:  change “albumens” to “albumen”  (remove the “s”)

Response 6: Done as requested (Please check Figure 1 in the RVMS). 

Point 7: Line 356:  change “used” to “using”

Response 7: Done as requested (Please check Figure 1 in the RVMS). 

Point 8: Line 379: change “consumed” to “consuming”

Response 8: Done as requested

Point 9: Lines 385-387:  Fix this sentence.  I think you mean to say something like this:

“The consumption of the white clover and chicory-clover mixture was more effective than either alone on the levels of MUFA and PUFA, which play a crucial role among unsaturated fatty acids, mainly due to their physiological functions.”

Response 9: Done as requested.

Point 10: Line 412-414:  Change “Indeed, PC1 indicate that FAs  from the TR and MIX groups had a high effect on the FA profiles of the egg yolks compared to the CI group.”  To “ Indeed, PC1 indicates that FAs from the TR and MIX groups had a greater effect on the FA profiles of the egg yolks compared to the CI group.

Response 10: Done as requested

Point 11: Line 415: Change  “because the low n-6/n-3 ratio”  to “because a low n-6/n-3 ratio”

Response 11: Done as requested

Point 12: Lines 439-440:  Change “These results can be beneficial for their own production system of individual poultry producers.”  To “Implementing this forage program within their own production system may be beneficial to individual poultry producers.”

Response 12: Done as requested

Round 2

Reviewer 2 Report

I would like to thank the authors for reviewing and accepting all the comments and suggestions. In my opinion the article is now acceptable for publication.

This manuscript is a resubmission of an earlier submission. The following is a list of the peer review reports and author responses from that submission.

Round 1

Reviewer 1 Report

My main issues concern the lack of some chemical analyses and statistical methods used in this study and .

The information about total fat content in yolk must be given. Showing the FA composition of basal fed (or at least primary fat source) is also required. without this data ,it is difficult to compare yolk results from hens of CD group with CD, TR or MIX.

PCA should be removed from the manuscript for many reasons. As a rule of thumb, the minimum sample size of at least 5 times to the number of variables is recommended for PCA. While some researchers suggested the sample size of at least 10 (or even 20) is required, I think that at minimum ratio of 5 would be sufficient for PCA analysis for FA data. In your data you have 11 variables (FA) and only 12 observation.  Were PCA assumptions checked (Bartlett's sphericity test and the KMO index (Kaiser-Mayer-Olkin))?  Also description of PCA results in some fragments does not make sense. In all quadrants of the PCA FA can be found, not only in three. Why there are four dots for CI? Also PCA separated only TR form CI and C groups for PC1.

Other comments:

Simples summary -is too technical, it is not “simple”

Introduction. A lot of contradictory information. For example, in L55-57 you state that there is no clear evidence that performance and egg quality are improved when hens are kept with access to areas with vegetation, and then in L59-61, L65-71, L75-77 you write about reported positive effects on performance and egg quality of hens raised in organic systems with herbs.

L77 References needed.

When you formulae a aim of the study, please do not write that the aim was to "test for distinct differences in the above-mentioned parameters ". Also research hypothesis is missing.

L98 unit is missing

L101 correct “work” to “period”

L103 did you measured egg production during pre-trail period to state that replicate pens were homogeneous in term of egg production?

L104-105 and L112-113 lighting regime is not described properly

L129 “each period” : at this moment it is not clear what period  - you mean 14-d-long or 5-wks-long intervals?

L133 ref [23] does not provide ANY information about experimental procedures, especially about braking strength measurements. Please describe experimental protocols in details.

L160 manufacturer and catalog number of this cholesterol kit is needed.

L164 Were cracked eggs data transformed (logit or angular (arcsine)) before analysis? Percentage or proportion data generally require transformation before analysis to normalize the distribution of residuals.

Table 1: chemical composition of forage species should begiven in separate table. in present form, it suggest the presentation of nutrient intake by hens form each group (basal diet corrected for HI).

Table 2: Table 2 belongs to MM section as it describes the FA content of feed additives being under investigation, not the results of the study.

Table 3,4: tables not clear. Please change to more-standardized layout:  in Table 3: initial bw, final bw, feed intake, herbage intake (HI), similarly in Table 4 (yolk (g/100g egg), albumen, etc.). Egg weight unit should be given in bracket.

Table 4: Traits like albumen, yolk or shape index were not defined in MM  section. Do they represent quantities like Haugh unit and albumen height?

Table 5: sum SFA, sum MUFA, sum PUFA. Denote n-3 and n-6 FA

L357 Why did you measure your cholesterol when it was assumed that there would be no difference between the treatments?

Reviewer 2 Report

-line 34-38 needs to report the numbers not only p values.

-line 56 remove the word worsend just say improved or not.

-line 73 change no studies to limited studies related to outdoor based system.

-line 101 mention the vaccine company.

-Overall p<0.05 needs to be written using the symbol of below or equal symbol. i couldn`t type it here but there is symbol indicates below or equal. 

Reviewer 3 Report

1. The research is interesting. 2. The FA contents in egg in Table 3 are not consistent with CI and TR in Table 5. The Polyunsaturated fatty acids in CI is higher that in TR, but it not found in egg. Why? 3. The yolk color in the CI, TR and Mix in table 4 is not higher than that in C. Why?

Reviewer 4 Report

This is a good study, and as the authors suggest, individual poultry producers need to determine how to use this information within their own production system.  I would suggest that if using CI and TR for pasture the producers might add extra vitamin D to the base diet and provide oyster shell free choice to help compensate for the egg shell issues.

The corrections I found are thus:

Line 21-22 change  "chicory (CI)- and white clover (TR)- "  to "chicory (CI)- or white clover (TR)-"

Line 33 change "chicory (CI)- and white clover (TR)-," to "chicory (CI)- or white clover (TR)-,"

Line 64  add a comma after the word carbohydrate." (carbohydrates, protein, vitamin ..."